# Predictions on the Phase Constitution of SmCo_7−X_M_x_ Alloys by Data Mining

**DOI:** 10.3390/nano12091452

**Published:** 2022-04-24

**Authors:** Guojing Xu, Hao Lu, Kai Guo, Fawei Tang, Xiaoyan Song

**Affiliations:** Faculty of Materials and Manufacturing, Key Laboratory of Advanced Functional Materials, Ministry of Education of China, Beijing University of Technology, Beijing 100124, China; gj.xu@emails.bjut.edu.cn (G.X.); kevin_gk@emails.bjut.edu.cn (K.G.); faweitang@bjut.edu.cn (F.T.)

**Keywords:** permanent magnets, machine learning, phase stability, composition design, grain size

## Abstract

Based on a home-built Sm-Co-based alloys database, this work proposes a support vector machine model to study the concurrent effects of element doping and microstructure scale on the phase constitution of SmCo_7_-based alloys. The results indicated that the doping element’s melting point and electronegativity difference with Co are the key features that affect the stability of the 1:7 *H* phase. High-throughput predictions on the phase constitution of SmCo_7_-based alloys with various characteristics were achieved. It was found that doping elements with electronegativity differences with Co that are smaller than 0.05 can significantly enhance 1:7 *H* phase stability in a broad range of grain sizes. When the electronegativity difference increases to 0.4, the phase stability becomes more dependent on the melting point of the doping element, the doping concentration, and the mean grain size of the alloy. The present data-driven method and the proposed rule for 1:7 *H* phase stabilization were confirmed by experiments. This work provides a quantitative strategy for composition design and tailoring grain size to achieve high stability of the 1:7 *H* phase in Sm-Co-based permanent magnets. The present method is applicable for evaluating the phase stability of a wide range of metastable alloys.

## 1. Introduction

Sm-Co-based permanent magnet alloys are the most promising permanent magnet materials that work at high temperatures, especially in an environment hotter than 500 °C. In traditional Sm-Co binary alloy systems, the SmCo_7_ compound possesses intermediate magnetic properties between those of the SmCo_5_ and Sm_2_Co_17_ compounds at room temperature. However, the SmCo_7_ compound has a much smaller temperature coefficient of coercivity (about −0.13%/°C) than that of the Sm_2_Co_17_-based magnets (approximately −0.3%/°C) [1]. As such, SmCo_7_-based magnets are recognized as promising candidates for high-temperature applications [2,3]. One of the big challenges in the development of SmCo_7_-based magnets is the stability of the single SmCo_7_ phase with a TbCu_7_-type (noted as 1:7 *H*; “*H*” represents “hexagonal”) structure. The 1:7 *H* structure is unstable at room temperature and is easily decomposed into Sm_2_Co_17_(R) and SmCo_5_(H) phases [4], which restricts the applications of SmCo_7_-based alloys. Therefore, the stabilization of the 1:7 *H* structure is crucial for the development of SmCo_7_-based alloys for high-performance applications.

At present, there are two main methods for stabilizing the TbCu_7_-type structure and obtaining the SmCo_7_-based alloys with a single metastable phase at room temperature. One is utilizing the nano-effect through the spark plasma sintering (SPS) technique. The grain size can be reduced to below the critical value predicted by the model calculations for the phase transition threshold [5,6]. The other approach that researchers employ to stabilize the metastable phase is element doping, and nearly 20 doping elements have been reported in SmCo_7_-based alloys [7,8,9,10,11,12,13,14,15,16,17,18,19,20,21,22,23,24,25]. Some of the doping elements, at an appropriate concentration, can cause the alloys to include the 1:7 *H* single phase. This doping method technique can enable the TbCu_7_-type phase at room temperature and improve the SmCo_7_-based alloy’s magnetocrystalline anisotropy.

In the past decade, many elements have been investigated as dopants to be used in the Sm-Co system in order to study their effects on the phase stability of the alloys; however, both computations and experiments have mostly been based on the conventional trial-and-error method. Moreover, most studies have focused on the rule that single variables affect the 1:7 *H* phase stability, whereas the cooperative effects of a doping element and grain size on the phase stability of Sm-Co-based alloys have rarely been studied. Therefore, it is difficult to identify the common law of phase stability from previous research or to obtain a guiding rule for the design of Sm-Co systems. First-principles and thermodynamic calculations for Sm-Co-based alloys are limited only to some specific systems and are difficult to carry out in large-scale calculations with high computational efficiency [26]. In contrast, with the rapid development of machine learning (ML) approaches and their applications in the field of materials design in recent years [27,28,29,30], data-driven methods exhibit an irreplaceable advantage for investigating the effects of multiple factors on material properties. However, attempts to apply machine learning methods to study and optimize the phase constitution of Sm-Co-based alloys have been limited in the literature.

In this study, we applied machine learning and feature screening methods to study the effects of doping elements on the phase constitution of SmCo_7−x_M_x_ alloys, based on the SmCo_7−x_M_x_ alloy phase constitution data set from our home-built Materials Genome Initiative (MGI) database of Sm-Co materials. The intrinsic features of doping elements determined the phase constitution of SmCo_7−x_M_x_ alloys. High-throughput predictions of SmCo_7−x_M_x_ alloy phase constitution with different grain sizes, doping elements, and doping amounts were achieved, and we proposed a universal rule of SmCo_7_-based alloy phase constitution under the concurrent effects of element doping and microstructure scale. This work provides a quantitative strategy for composition design and tailoring grain size to achieve high stability of the 1:7 *H* phase in Sm-Co-based permanent magnets. The present method is applicable for evaluating the phase stability of a wide range of metastable alloys.

## 2. Materials and Methods

### 2.1. Data Set Used in the Analysis

The data set used in this work was extracted from our home-built Materials Genome Initiative (MGI) database of Sm-Co materials, which contained more than 1300 Sm-Co-based alloys. This database is a highly structured Sm-Co-based materials database with the most abundant data content compared with other Sm-Co-based materials databases. Each data item includes the process, performance, composition, phase constitution, etc., of the material in question.

A total of 508 binary SmCo_7_ and ternary SmCo_7−x_M_x_ data items [7,9,10,11,13,14,18,19,20,21,22,31,32,33,34,35,36,37,38,39,40,41,42,43,44,45,46,47,48,49] with integrated alloy composition and phase constitution were screened to construct a “SmCo_7−x_M_x_ alloy phase constitution data set”, abbreviated herein as “data set”. The data set used for ML analysis contains 263 cases with the 1:7 *H* single phase and 245 cases with multiphases.

### 2.2. Feature Construction and Data Pre-Analysis

According to the literature, the potential factors affecting the phase constitution mainly include the following. The meaning of each symbol representing the potential factors is indicated in Appendix A.

Firstly, the alloy composition features include the kind of doping element (*M*) and its proportion (*x_c_*). *x_c_* represents the percentage of the doping element substituting Co ($x_{c}=100x/7$).

Secondly, the form features ($X_{form}$) include ribbon, bulk, and powder.

Thirdly, the process features consist of the final step in commonly used Sm-Co-based alloy preparation processes ($X_{proc}$), including spark plasma sintering (SPS), melting, spinning, and annealing. Although there is a lack of detailed process parameters in the literature generally, processing can be distinguished by the preparation technique combined with other features (such as the average grain size) [12]. Due to an imbalance in the data volume regarding different preparation processes, the processes can be further divided into two categories ($X_{proc}^{*}$): as-prepared processes and heat treatment:(1)$$X_{proc}^{*}=\{\begin{array}{c} \mathrm{Preparation},\;X_{proc}=\mathrm{SPS},\mathrm{Spinning},\mathrm{Melting}\\ \mathrm{Heat}\;\mathrm{treatment},X_{proc}=\mathrm{Annealing} \end{array}$$

The further classification of processes does not result in a loss of alloy information. The combination of form features and process features can define the course of Sm-Co-based alloy preparation. For example, the alloys in bulk forms can be prepared by SPS, melting, or annealing, and these bulk alloys are distinguished by their process features.

Finally, the average grain size of Sm-Co-based alloys (*d*) is also an important feature. Due to the different processes, the grain size of the material varies from 10 nm to 5000 nm. This large grain size range can cause the ML model to be less sensitive to grain size, resulting in the model not being able to accurately predict the phase stability of the alloys with small grain sizes. To avoid the adverse effects of a large grain size range on the machine learning model, we took the logarithm of the grain size, which is denoted as *lgd*.

Phase constitution (*G*(1:7 *H*)) is the target variable of this study. According to the commonly used description for phase constitution investigations, the phase constitution can be semi-quantitatively expressed as five categories of ‘*All*’, ‘*Main*’, ‘*Part*’, ‘*Minor*’, and ‘*None*’.

Due to the small volume of data collected, we further considered whether the SmCo_7−x_M_x_ alloy is composed of a single 1:7 *H* phase as the target variable, and converted the multi-classification problem into a binary classification problem, which is denoted as *F*(1:7 *H*):(2)$$F\left(1:7\;H\right)=\{\begin{array}{c} All,G\left(1:7\;H\right)=All\\ \mathrm{Other},G\left(1:7\;H\right)=Main,Part,Minor,None \end{array}$$

In other words, whether the SmCo_7−x_M_x_ alloy is a 1:7 *H* single phase can be expressed as the following:(3)$$F\left(1:7\;H\right)\sim \left(M,x_{c},X^{*}{}_{proc},X_{form},lgd\right)$$

After categorizing the target variables and features, we made a preliminary analysis of the available data, as shown in Figure 1. The volume of data from different doping elements is not balanced. The doping elements that researchers focus on are mainly transition metals and IIIA and IVA group elements. Two continuous numerical variables, i.e., *lgd* and *x_c_*, are used for multi-dimensional visualization, as shown in Figure 1. Under the synergistic effects of grain size and doping amount, the 1:7 *H* single phase appears in different regions on the map. The ternary alloy exhibits nano-effects similar to the binary SmCo_7_ to some extent, and the 1:7 *H* phase tends to be stable in the alloys with small grain sizes, such as the alloys with Ga and C doping. At the same time, the doping of Si shows a good ability to stabilize the 1:7 *H* phase, as shown by the experimental data in Figure 1a.

### 2.3. Machine Learning (ML) Algorithms

A machine learning approach was adopted to study phase stability. Before model training, the categorical features ($X^{*}{}_{proc},\;X_{form},\;F\left(1:7\;H\right)$) were numerically transformed into dummy features, and the numerical features were normalized. The features with high correlations were filtered and removed. A support vector machine with a radial basis function kernel (SVM_rbf) [50] machine learning model was used for model training and feature selection [51]. In the feature selection process, Area Under Curve (AUC) values [52] on the validation set were used as an indicator to evaluate the influence of the feature subsets on phase stability, and the accuracy rate was chosen to evaluate the prediction error on two cross-validation methods. The Random Cross-Validation (RCV) and Leave Two Elements Cross-Validation (LTECV) methods were used in the model training for the hyperparameters [53].

### 2.4. Feature Engineering

#### 2.4.1. Feature Correlation Analysis

Elements can be distinguished by their intrinsic characteristics, such as atomic number, electronegativity, conductivity, etc. The difference between elements can essentially be expressed as the difference between these intrinsic properties. The common intrinsic properties of elements were used to numericalize the elements in this study, and a total of 14 candidate features, as shown in Table 1, were selected. The intrinsic properties of elements are denoted as $X_{M}$.

It should be noted that elemental intrinsic properties may be interdependent, which has a negative impact on ML predictions. Therefore, it is necessary to reduce the dimensionality of the highly correlated variables and remove the features with strong linear correlations. The Pearson correlation coefficient analysis was used for these intrinsic properties to screen the representative features with weak collinearity, which is defined as:(4)$$\rho \left(x_{1},x_{2}\right)=\frac{cov\left(x_{1},x_{2}\right)}{\sigma_{x1}\sigma_{x2}}$$
where *ρ* is the Pearson’s coefficient, $x_{1}$ and $x_{2}$ are two elemental intrinsic properties, *cov* is the covariance function, and *σ* represents the standard deviation of the elemental intrinsic properties. It is generally agreed that there is multicollinearity between features when the correlation coefficient is greater than 0.8, and only one of these two features is needed for further analysis in ML. Figure 2 shows a comprehensive information map of the 14 intrinsic properties of elements. Atomic number (*Z*) with standard atomic weight ($A_{r}$); melting point ($T_{m}$) with boiling point ($T_{b}$) and heat of vaporization ($\Delta H_{vap}$); and electrical conductivity (κ) with thermal conductivity (λ) have significant multicollinearity. Among these properties, standard atomic weight ($A_{r}$) is a more objective intrinsic property than atomic number (*Z*), and melting point ($T_{m}$) and electrical conductivity (κ) are more commonly used in ML studies. In addition, electron volume ($V_{a}$) yields a cubic relationship with the atomic radius ($r_{a}$), and work function (φ) and electron density ($N_{WS}$) also show a strong correlation with atomic radius ($r_{a}$). Therefore, seven intrinsic properties, including atomic radius ($r_{a}$), standard atomic weight ($A_{r}$), heat of fusion ($\Delta H_{f}$), melting point ($T_{m}$), electrical conductivity (κ), the first ionization energy ($E_{i,\;1st}$), and electronegativity (χ) were chosen to express the element characteristics, namely as:(5)$$\;X_{M}\sim \left(r_{a},A_{r},\Delta H_{f},T_{m},\kappa ,E_{i,1st},\chi \right)$$

#### 2.4.2. Evaluation for the Effect of Intrinsic Features on the Stability of the 1:7 *H* Phase

Further, new intrinsic features were constructed to expand the feature candidate space. Considering the amount of doping and the interactions between doping elements in SmCo_7−x_M_x_ alloys with Sm and Co, new functions were constructed, as shown in Table 2. Based on the functions in Table 2, 7 × 3 = 21 candidate intrinsic features were created. To attain a reliable ML model for phase constitution prediction, it was necessary to further optimize its intrinsic feature space subsets.

A partial exhaustion method was applied to create subsets of the intrinsic feature space. Because the three new features constructed in Table 2 belong to one intrinsic elemental property, these three intrinsic features cannot be in the same subset of features. Therefore, the subset construction was divided into two steps to avoid features from the same intrinsic elemental property being placed into one subset: the first step was selecting *n* features from the 7 features in Equation (5); the second step was choosing any one of the three constructors for each feature. Therefore, the total number of all the possible subsets of features is:(6)$$\overset{7}{\underset{n=0}{\sum }}C_{7}^{n}\times 3^{n}$$
*lgd*, $X_{form}$, and $X^{*}{}_{proc}$ were used as the bottom features (BF), which are involved in every ML training with the possible subsets of features. The feature selection process used 80% of the data set as the training set and 20% as the validation set. SVM.rbf was used to train all feature subsets, and the combinations of *σ* = 0.1, 0.4, and *C* = 1, 4 were applied as hyperparameters in order to reduce the error caused by a single hyperparameter. All feature subsets were trained under the four hyperparameter combinations to obtain an average AUC value. The closer AUC was to 1, the higher the model accuracy was. Figure 3 shows the mean AUC values of all feature subsets. According to Equation (6), 16,384 feature subsets were involved in the ML training. Due to the huge number of subsets, it was unnecessary to count the mean AUC corresponding to each feature subset. Therefore, the mean values of AUC of the optimal feature subsets with different numbers of intrinsic features were analyzed to select the most important intrinsic features affecting the 1:7 *H* phase stability. When *n* = 0, only the bottom features were involved in the subset, and the mean AUC obtained by the training was 0.65. After introducing one intrinsic feature (*n* = 1), the mean AUC of all feature subsets increased significantly, and the highest mean AUC of 0.75 was obtained from the subset of BF and $x_{sub}\cdot \left|T_{m}{}_{Co}-T_{m}{}_{M}\right|$. After introducing two intrinsic features (*n* = 2), a total of $C_{7}^{2}\times 3^{2}=189$ feature subsets were involved in the training, and the best performing subset was the combination of BF, $x_{sub}\cdot T_{m}$ and $x_{sub}\cdot \left|\chi_{Co}-\chi_{M}\right|$, improving the mean AUC to 0.78. When *n* = 3,4,5, the best mean AUC were 0.79, 0.80, and 0.80, and the AUC decreased after *n* > 5. The mean AUC with different numbers of features and the optimal features are shown in Appendix A. It can be seen from Figure 3 that the AUC of the model can be improved by increasing the number of intrinsic features when *n* < 5; however, the growth rate in the AUC is very limited after *n* > 2. Considering the finite amount of data, and to prevent overfitting, it is believed that the inclusion of a third intrinsic feature is not necessary in the present study. Using ML and feature engineering, we found that $x_{sub}\cdot T_{m}$ and $x_{sub}\cdot \left|\chi_{Co}-\chi_{M}\right|$ were the two most important intrinsic features that affect the phase constitution of SmCo_7−x_M_x_ alloys.

## 3. Results and Discussion

The six features that most strongly affect the phase constitution of SmCo_7−x_M_x_ alloys were selected by feature engineering, and the relationship for ML training can be expressed as:(7)$$F\left(1:7H\right)\sim \left(T_{m},\;\left|\chi_{Co}-\chi_{M}\right|,\;x_{sub},\;\;X^{*}{}_{proc},\;X_{form},\;lgd\right)$$

According to the selected features and the machine learning algorithms, a high-throughput prediction for the phase constitution of SmCo_7−x_M_x_ with a doping element of M can be achieved. Virtual samples were constructed according to the characteristics of the descriptors, and the design rules are shown in Appendix A. The selection of doping elements was based on the fact that transition metal and IIIA and IVA elements are often used in experiments. It resulted in 1,543,500 alloys in total, with 35 kinds of doping elements.

### 3.1. A More Suitable Cross-Validation Method Developed for Predicting “Unknown Elements”

Traditional machine learning strategies usually use random cross-validation (RCV) methods to avoid over-fitting in the model, and this method reflects the predictive ability of randomly generated samples in the explored space. However, it cannot well predict the “new elements”. Liu [54] used Leave One Doping Element Out Cross-Validation (LEOCV) to predict the effects of doping elements on the saturation magnetization of Sm-Co-based ternary alloys and achieved good prediction ability. However, for the classification problem of the phase constitution investigation, the LEOCV method cannot perform well enough. That is because the data volumes for various doping elements in the data set differ significantly, which may cause an individual element with a small amount of data to be overweighted in the accuracy calculations. Moreover, the phase constitutions of ternary compounds with some specific elements are all 1:7 *H* single phase (Si, Al, and Fe) or multiphases (Cr), which would result in a serious imbalance in the grouping of target variables and hence the inaccuracy of the LEOCV method.

Therefore, we developed a Leave Two Elements Cross-Validation (LTECV) method to further process the data. The doping elements were first sorted by the volume of data in the data set, and then we considered the ratio of single-phase and multiphase data for each element to group two elements together to balance the amount and ratio of the data. The grouping results are shown in Appendix A, and a total of 10 groups of elements were generated. Nine groups of elements were selected as the training set, and one group was chosen as the validation set for cross-validation. Correspondingly, the RCV method used ten-fold cross-validation for comparison. The “data set” was randomly divided into 10 groups, of which 9 groups were the training set, and 1 group was the validation set for cross-validation. It should be noted that the number of samples used may be different for these two CV methods. The RCV method tries to equalize the 10 groups of the “data set”. However, the LTECV method divides the “data set” into groups by elements, and the amount of data in each group is not the same (as shown in Appendix A). Thus, it was difficult to ensure that the sample size in each data group for the two CV methods was exactly the same. The effect of different sample sizes on model accuracy is discussed later. SVM.rbf was used for training, and the accuracy rate was chosen to evaluate the prediction error. We recorded the mean accuracy of the model on the test set under a specific hyperparameter and the confusion matrix of the optimal prediction results from the two cross-validation methods, as shown in Figure 4. The optimal hyperparameter combination was *C* = 2^21.5^, *σ* = 2^−3.5^ in the RCV method, and the combination in the LTECV method was *C* = 2^19.5^ and *σ* = 2^−5.8^, which is similar to that of the RCV method. 

From the confusion matrix of the two CV methods with optimal hyperparameter combinations (Figure 4a,c), it can be seen that the difference in sample size between the two methods was not large (LTECV:RCV = 43:50). Additionally, the ratio of single-phase and multiphase cases was also similar for these two CV methods (23:20 for LTECV and 26:24 for RCV). For a data set with balanced data distribution, it is believed that the accuracy rate can be used to evaluate the model. The best mean accuracy rates were 0.88 and 0.77 in the two CV methods, respectively, and the accuracy for RCV was higher than for LTECV.

To evaluate the generalization ability of the model, the two cross-validation methods were used to train the model and predict the virtual samples doped with Mn. The experimental data for the prepared bulk SmCo_7−x_Mn_x_ (x = 0.1, 0.3, 0.5, 0.7, 1) alloys [25], which did not participate in the training process, were used as the test set. The prediction and the experimental results were compared to evaluate the generalization ability of different methods. The results are shown in Figure 5. It can be seen that, although RCV performs better on the validation set, it is more prone to overfit in a large virtual sample space. However, our proposed LTECV method shows a stronger generalization ability, and the predicted phase constitution on the test set is highly consistent with the experimental results. This method is more applicable to predict “new elements” compared with the traditional RCV method.

### 3.2. High-Throughput Prediction for the Phase Constitution of Sm-Co-Based Alloys

The trained model was used to predict the phase constitution of a large number of SmCo_7−x_M_x_ alloys. More than 1.5 million virtual samples were predicted. The model can analyze the influence of the preparation process, material form, doping elements, doping amount, and grain size on the alloy phase constitution. It found that prepared powdery alloys were difficult to crystallize, so the prediction results had little theoretical importance in the analysis. The number of data items with a single phase in the bulk and ribbon samples after heat treatment was significantly reduced, which is consistent with reports in the literature. Therefore, the prediction results for prepared (SPSed and melted) bulk alloys were plotted on the periodic table to analyze the phase constitution law under the comprehensive effect of doping elements, doping amount, and grain size, to guide the design of SmCo_7_-based alloys with a single 1:7 *H* phase.

According to the predicted results in Figure 6, it can be seen that the doping elements can be clearly classified into three categories based on their effect on phase stability. The first category is elements that can stabilize the 1:7 *H* phase in a large grain size range, such as Fe, Ni, Cu, Ag, Si, and Tc. The second includes elements that have a non-monotonic trend in the regulation of 1:7 *H* phase stability by the grain size and doping amount, such as Ti, V, Nb, Mo, and Ta. The largest critical grain size for phase decomposition in alloys occurs when the doping amount is between 0.5–0.7. The stabilizing effect of doping on the 1:7 *H* single phase gradually weakens if the amount of doping continuously increases, and the critical grain size that can maintain phase stability gradually decreases. If the doping amount is too large, the stable 1:7 *H* phase cannot exist, no matter how small the grain size is. Additionally, the last category includes those elements that can only maintain the stable existence of the 1:7 *H* single phase in a narrow doping amount and grain size range, such as In and Hf. 

Such differences arise from the intrinsic features of the doping elements. In order to quantitatively analyze the effect of dopant elements on phase stability, the doped elements were plotted on the plane of the two features selected in Section 2.4.2, as shown in Figure 7. It shows that if $\left|\chi_{Co}-\chi_{M}\right|$ < 0.05, the doping elements can significantly promote the formation of the 1:7 *H* single phase. With the increase in $\left|\chi_{Co}-\chi_{M}\right|$, the melting point of doping elements plays a significant role in the phase stability. When 0.05 < $\left|\chi_{Co}-\chi_{M}\right|$ < 0.40, doping with high melting point elements results in the maximum critical grain size for the phase decomposition of SmCo_7_-based alloys at a doping amount between 0.5 and 0.7. However, if low melting point elements such as In and Cd are doped, they can only stabilize the 1:7 *H* phase in a small range of grain size and doping amount. Once $\left|\chi_{Co}-\chi_{M}\right|$ > 0.40, these elements, such as Zr, can only maintain the stable existence of the 1:7 *H* single phase in a narrow doping amount and grain size range. 

According to the predictions shown in Figure 6, three representative elements that are suitable for stabilizing the 1:7 *H* phase were selected to experimentally verify the prediction accuracy of the ML model. As shown in Figure 8, the experimentally as-prepared SmCo_7−x_Si_x_ (x = 0, 0.4, 0.6, 0.9), SmCo_7−x_Ni_x_ (x = 0.1, 0.5), and SmCo_6.5_Fe_0.5_ bulk alloys from SPS can maintain the 1:7 *H* phase stable in the grain size range of 25–44 nm. These experimental results are highly consistent with the prediction from the ML model, as shown in Figure 8c. In addition, all the as-cast SmCo_7−x_Si_x_ (x = 0, 0.4, 0.6, 0.9) ingots with large grain sizes and doped with Si crystallize in a hexagonal TbCu_7_-type structure [21]. This indicates that the addition of Si can stabilize the 1:7 *H* phase in a large grain size range, and Si, Fe, and Ni can promote the formation of the 1:7 *H* single phase. As for the elements that can only maintain the stable existence of the 1:7 *H* single phase in a narrow doping amount and grain size range, the predictions for these elements can be verified by the data in the literature. For instance, the Hf element can only stabilize the 1:7 *H* phase with a doping amount of less than 0.25 and with a grain size range of 20–30 nm [24]. The ML strategy proposed in this study can realize high-throughput and high-precision predictions for the phase constitution of as-prepared SmCo_7−x_M_x_ bulk alloys.

In previous reports, most researchers only studied the effect of doping amounts on the phase stability of alloys with one or a few kinds of doping elements. In addition, some other studies investigated the role of grain size on the stabilization of the 1:7 *H* phase in Sm-Co-based alloys. In our study, the cooperative effects of grain size and doping amount on the phase stability were quantitatively investigated for the first time, scanning the periodic table of elements. The effect of elemental intrinsic features on the phase stability of the 1:7 *H* phase in Sm-Co-based alloys was determined based on feature engineering in the machine learning approach.

Since this paper makes high-throughput predictions for the effects of doping elements and grain size on the phase stability of the 1:7 *H* phase over the entire periodic table of elements, it is not possible to experimentally verify all the predictions in one study. Our predictions provide guidance for subsequent experiments and need further verification. In addition, machine-learning-based methods and feature engineering usually lack deep physical foundations; thus, further studies combined with theoretical models may be required in the future.

## 4. Conclusions

In the present work, the phase constitutions of Sm-Co-based alloys with doping elements were predicted by an ML method. It revealed that the stability of the 1:7 *H* metastable phase in the doped alloys was determined by the comprehensive effect of the grain size, doping amount, and the intrinsic properties of the doping elements. The elements that can promote the formation of the 1:7 *H* single phase in Sm-Co-based alloys were provided, and a regulating approach for 1:7 *H* phase stability was proposed. The main conclusions drawn are as follows.

(1) A support vector machine learning model was established to predict the phase constitution of SmCo_7−x_M_x_ alloys. Based on the home-built MGI database of Sm-Co materials, a data set containing 508 Sm-Co binary and ternary alloy samples was constructed. Combined with feature engineering and machine learning, an exhaustive method was applied to find that the melting point of the doping elements and the electronegativity difference with Co were the two key features that affected the phase constitution of SmCo_7−x_M_x_ alloys.

(2) The LTECV method was proposed to enhance the generalization ability of the model. Compared with RCV, the optimized machine learning model has a better generalization ability and can effectively avoid overfitting. This method can predict “unknown elements” using existing data.

(3) High-throughput prediction for the phase constitution of as-prepared SmCo_7−x_M_x_ nanocrystalline bulk alloys was achieved using the doping elements, the doping amount, and the mean grain size as key variables. It was revealed that Fe, Ni, Cu, Si, Ag, and Tc have a small electronegativity difference with Co and can significantly promote the formation of the 1:7 *H* single phase. By regulating the doping amount, these elements can maintain the stable existence of the 1:7 *H* single phase in a large grain size range.

(4) The predictions from the ML strategy were verified by the experimentally prepared SmCo_7_-based alloys doped with Ni, Fe, and Si. The theoretical model and method established in this paper provide quantitative guidance and a scientific basis for using composition design and grain size tailoring to achieve high 1:7 *H* phase stability in the SmCo_7−x_M_x_ alloy system.

## Figures and Tables

**Figure 1 nanomaterials-12-01452-f001:**
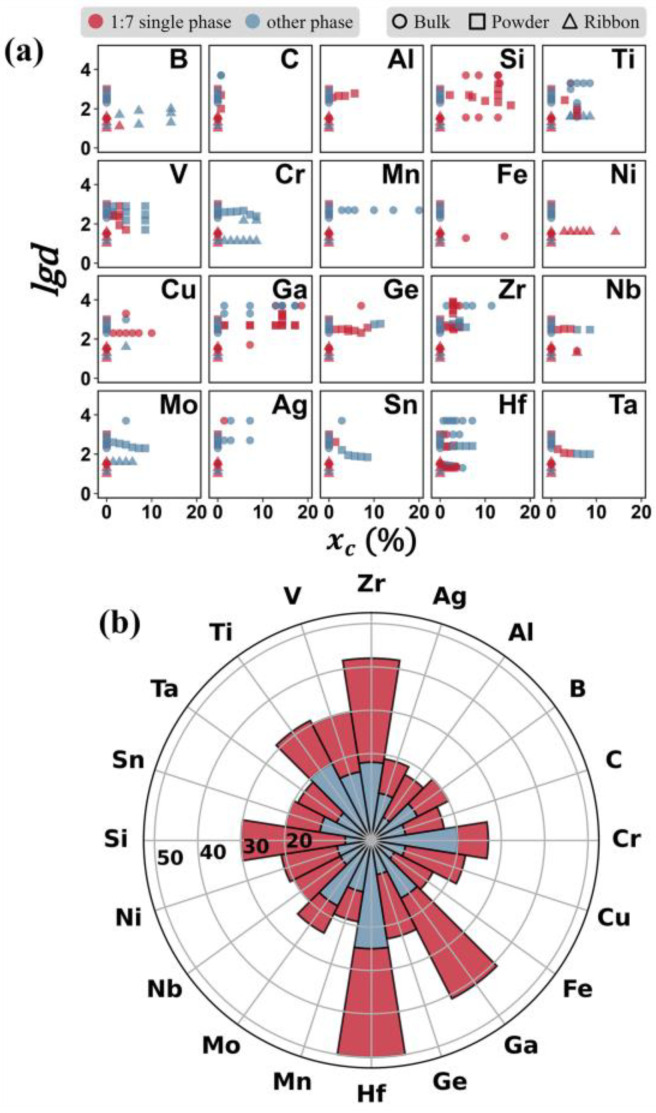
**Analyses of the data regarding phase stability reported in the literature for Sm-Co-based alloys with various doping elements.** (**a**) Data visualization and classification for the phase constitution of SmCo_7−x_M_x_ alloys under the comprehensive influence of the doping element, doping amount, and grain size. Different shapes represent different material forms. The *x*-axis is the doping amount of the elements, and the *y*-axis is the logarithmic of the grain size. (**b**) Number of samples with different doping elements in the data set. The red area indicates the number of samples with a single 1:7 *H* phase, and the blue area indicates the number of samples with multiphases.

**Figure 2 nanomaterials-12-01452-f002:**
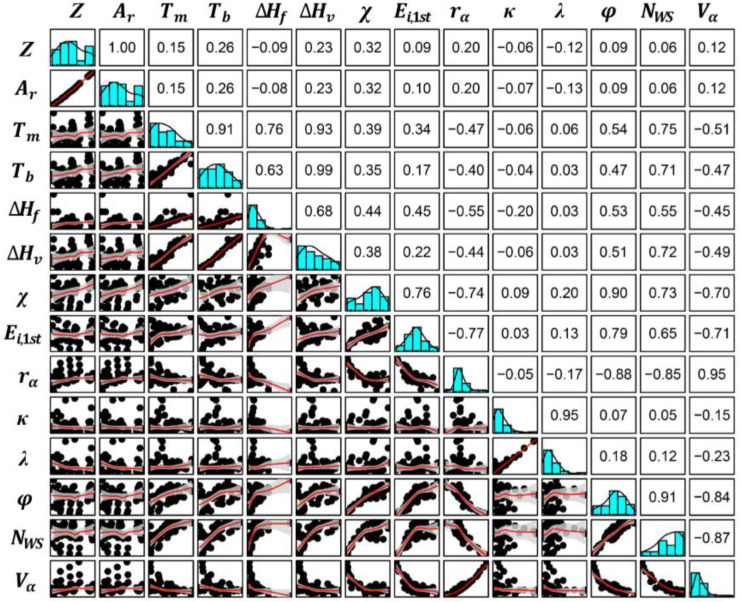
**Correlation analysis for different intrinsic properties of elements.** The diagonal squares in the matrix represent the distribution histogram of each feature. The bottom left shows the linear relationship between the two features, and the upper right numbers represent the Pearson correlation coefficients.

**Figure 3 nanomaterials-12-01452-f003:**
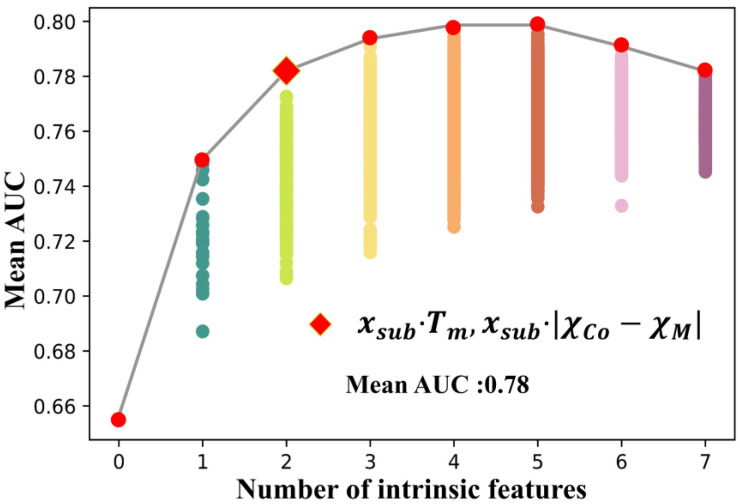
**The mean AUC values of all feature subsets**. The dots of different colors indicate the mean AUC corresponding to the subset with various numbers of features. The scattered points overlap with each other due to the large number of subsets with high numbers of features. The red points indicate the mean AUC of optimal feature subsets for different numbers, and the diamond represents the intrinsic features that we finally chose for machine learning.

**Figure 4 nanomaterials-12-01452-f004:**
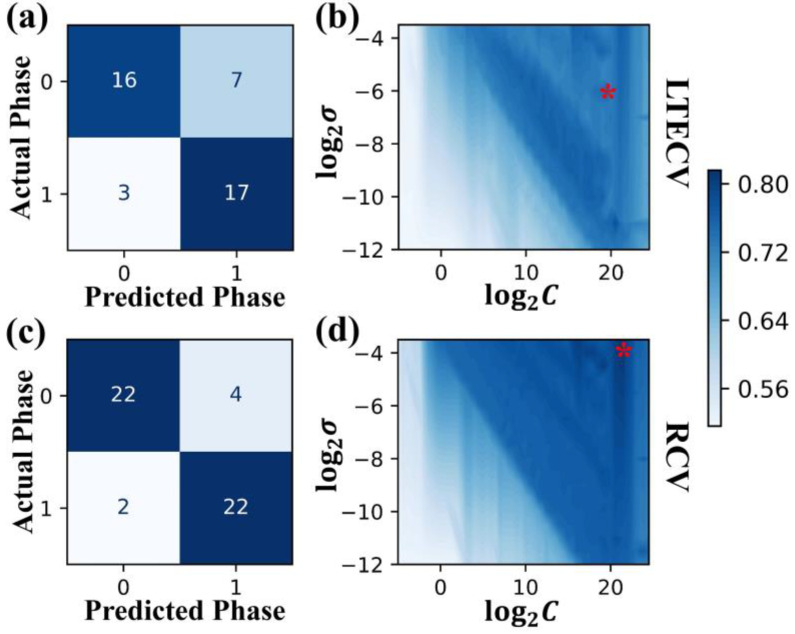
**Comparison of classification results for LTECV and RCV.** (**a**,**c**) The confusion matrix of the model trained by LTECV and RCV on the validation set, where 0 represents the single 1:7 *H* phase and 1 represents multiphases; (**b**,**d**) The mean classification accuracy rates of LTECV and RCV on the validation set under different hyperparameters, and the red star indicates the best hyperparameter combination.

**Figure 5 nanomaterials-12-01452-f005:**
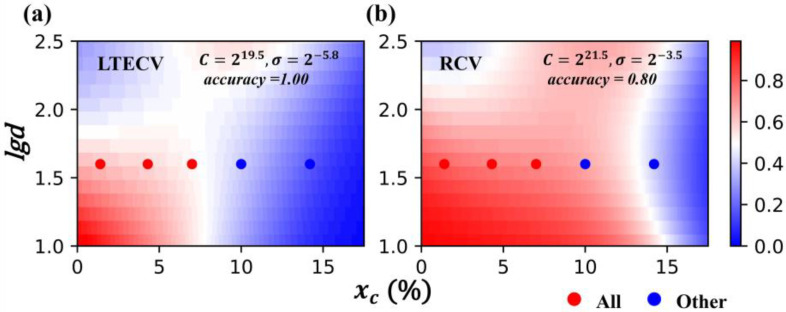
**Comparison of the generalization ability of two CV methods.** The color bar indicates the probability of generating the 1:7 *H* single phase, and red means it is easier to obtain the stable 1:7 *H* phase. Red points represent the test set, indicating the experimental data of prepared SmCo_7−x_Mn_x_ (x = 0.1, 0.3, 0.5, 0.7, 1) bulk alloys without training in the ML process [25]. (**a**) Experimental results and prediction results from LTECV; (**b**) Experimental results and prediction results from RCV.

**Figure 6 nanomaterials-12-01452-f006:**
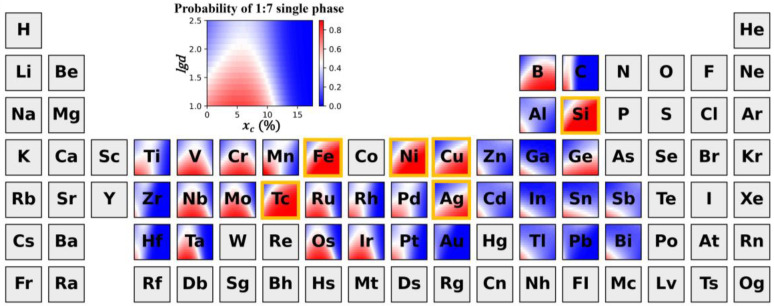
**High-throughput prediction for the phase constitution of prepared SmCo_7−x_M_x_ bulk alloys under the cooperative effect of composition and grain size.** The illustration above the periodic table is a guide bar to show the details. Color bar indicates the probability of generating the 1:7 H single phase, and red means it is easier to obtain the stable 1:7 H phase. The yellow boxes represent the “preferred elements” for stabilizing the 1:7 *H* phase.

**Figure 7 nanomaterials-12-01452-f007:**
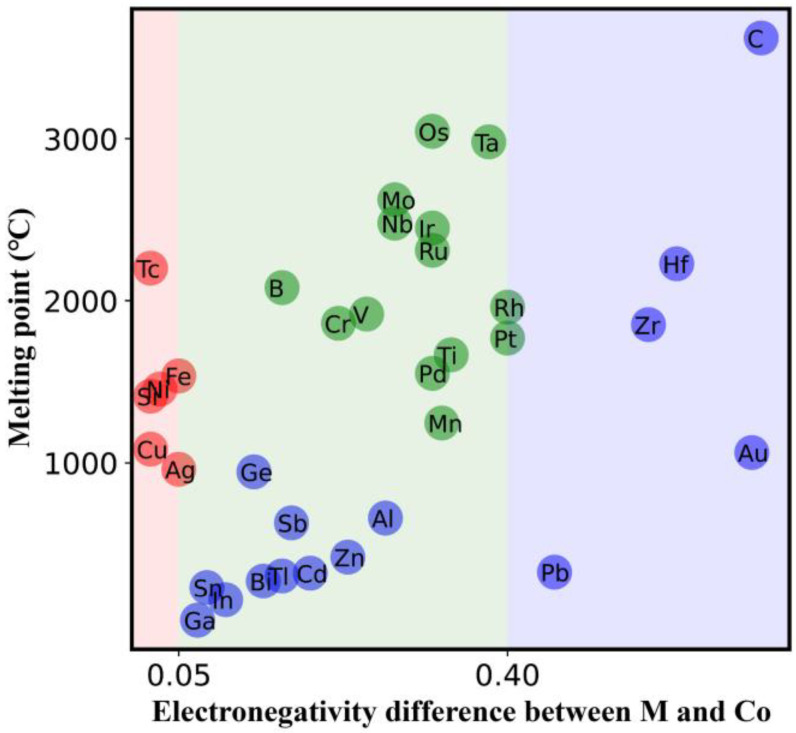
**Classification of the doping elements based on their **$T_{m}$**and**$\left|\chi_{Co}-\chi_{M}\right|$**.** The elements in red can promote the formation of the 1:7 *H* single phase, and the elements in blue can only stabilize the 1:7 *H* phase in a narrow doping amount and grain size range. The doping elements in green have the largest critical grain size for phase decomposition at doping amounts between 0.5 and 0.7.

**Figure 8 nanomaterials-12-01452-f008:**
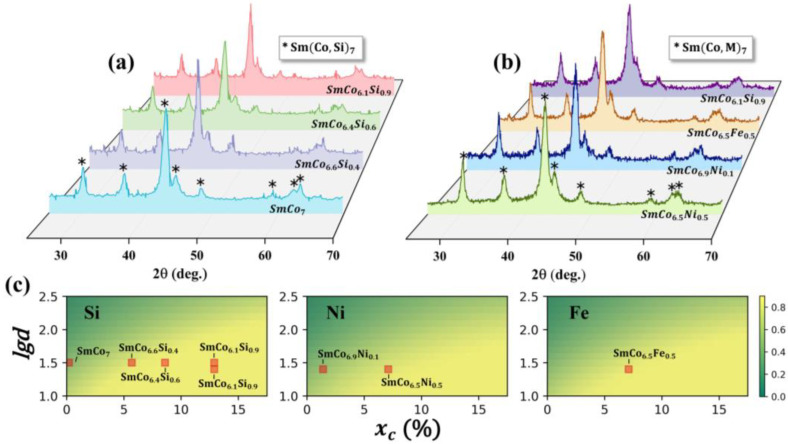
**Experimental verification for doping elements that can stabilize the 1:7 *H* single phase obtained by ML**. The asterisk represents the peak of the 1:7 *H* phase measured by XRD: (**a**) Phase analysis for SmCo_7−x_Si_x_ (x = 0, 0.4, 0.6, 0.9), and the mean grain size is 28–44 nm [21]; (**b**) Phase analysis for SmCo_6.5_Ni_0.5_, SmCo_6.9_Ni_0.1_, SmCo_6.5_Fe_0.5_, and SmCo_6.1_Si_0.9_, and the mean grain size is 25–30 nm; (**c**) Comparison of experimental results and ML predictions of SmCo_7−x_M_x_ (M = Si, Ni, Fe) alloys.

**Table 1 nanomaterials-12-01452-t001:** Intrinsic properties of elements and symbolic representation.

Name	Symbol	Feature Selection
Atomic number	*Z*	
Atomic radius	$r_{a}$	√
The first ionization energy	$E_{i,\;1st}$	√
Standard atomic weight	$A_{r}$	√
Melting point	$T_{m}$	√
Boiling point	$T_{b}$	
Electronegativity	χ	√
Electrical conductivity	κ	√
Heat of fusion	$\Delta H_{f}$	√
Heat of vaporization	$\Delta H_{v}$	
Thermal conductivity	λ	
Work function	φ	
Electron density	$N_{WS}$	
Electron volume	$V_{a}$	

**Table 2 nanomaterials-12-01452-t002:** New intrinsic feature constructor.

Function	Implication
$x_{sub}\cdot X_{M}$	The product of doping amount and element features
$x_{sub}\cdot \;|X_{Co}-X_{M}|$	Interaction between doping element and Co
$x_{sub}\cdot |X_{Sm}-X_{M}$ *|*	Interaction between doping element and Sm

$x_{sub}$ represents the proportion of doping element substituting Co ($x_{sub}=x/7$).

## Data Availability

The data presented in this study are available on request from the corresponding author.

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
