# Peer review of "Predictions on the Phase Constitution of SmCo7−XMx Alloys by Data Mining"

_nanomaterials, 2022, doi:10.3390/nano12091452_

Round 1

Reviewer 1 Report

The topic of the paper is interesting and very up-to-date. The literature basis is good and the research methodology is well described. The paper is well written and readable.

The paper itself has some flaws worth improving:

  • Please describe the research gap of the paper.
  • Describe the research questions.
  • Please describe the links between the research gap and the goal of the paper and research question. Write why the paper is important. What is the main contribution of the paper to the field?
  • Please add theoretical part of the paper with the extensive analysis of the theoretical basis.
  • The paper lacks discussion part. In this part Authors should wrote how they research are with competition to others researchers.
  • It would be good to add the limitation of the study.

Reviewer 3 Report

This a convincing example of a useful application of machine learning approaches to materials science. The methods employed are well described, and the results are solid.

I suggest publication of the manuscript  in Nanomaterials in its present form, except from a few typos or strange notation, for instance lines 228-230.

Round 2

Reviewer 1 Report

Authors have implemented my remarks.

Reviewer 2 Report

The authors' responses and revisions have satisfactorily addressed my comments on the previous version of the manuscript.  I recommend publication in its current version.